# The Influence of Conventional and Innovative Rehabilitation Methods on Brain Plasticity Induction in Patients with Multiple Sclerosis

**DOI:** 10.3390/jcm12051880

**Published:** 2023-02-27

**Authors:** Marta Milewska-Jędrzejczak, Andrzej Głąbiński

**Affiliations:** Department of Neurology and Stroke, Medical University of Lodz, ul. Zeromskiego 113, 90-549 Lodz, Poland

**Keywords:** brain plasticity, multiple sclerosis, virtual reality, rehabilitation, neuroplasticity biomarkers, neurorehabilitation, BDNF

## Abstract

Physical rehabilitation and physical activity are known non-pharmacological methods of treating multiple sclerosis. Both lead to an improvement in physical fitness in patients with movement deficits while improving cognitive function and coordination. These changes occur through the induction of brain plasticity. This review presents the basics of the induction of brain plasticity in response to physical rehabilitation. It also analyzes the latest literature evaluating the impact of traditional physical rehabilitation methods, as well as innovative virtual reality-based rehabilitation methods, on the induction of brain plasticity in patients with multiple sclerosis.

## 1. Multiple Sclerosis and Physical Rehabilitation

Multiple sclerosis (MS) is a chronic autoimmune disease of the central nervous system of unexplained etiology. The disease most often affects young adults, leading to developmental issues, such as paresis of the limbs, ataxia, visual disturbances, and sphincter disturbances [1]. In addition, patients with MS, including those with clinically isolated syndrome (CIS), demonstrate the deterioration of cognitive functions from the initial stages of the disease [2]. All of the mentioned symptoms lead to a decrease in the quality of life of MS patients and often to the need to give up work and personal life [3].

Despite many studies, the exact etiology of multiple sclerosis is still not fully known. It is believed that the development of the disease is caused by abnormal activation of the immune system induced by external factors in genetically predisposed individuals [4]. Since demyelination is a key feature of MS neuropathology, it has been hypothesized that antigens derived from myelin protein such as myelin basic protein (MBP), myelin-associated glycoprotein (MAG), myelin proteolipid protein (PLP), and myelin and oligodendrocyte glycoprotein (MOG) are the primary target of attack of the immune system. These antigens are recognized by periphery-activated CD4+ T cells, which use adhesion molecules and chemokines to cross the blood–brain barrier (BBB) and lead to a cascade of pro-inflammatory processes. Then, there is an increased cellular response, an influx of CD8+ T lymphocytes, as well as monocytes, NK cells, and neutrophils, and as a consequence, the formation of demyelinating and axonal damage [5]. It has also been shown that MS patients have an increased number of autoreactive CD4+ T cells in relation to the above-mentioned proteins. The pathogenesis of MS is also influenced by the humoral response with the pathogenic role of B lymphocytes. By secreting cytokines, B lymphocytes play a role in pro-inflammatory and regulatory processes. Additionally, via plasma cells, they play a role in the production of antibodies, including CSF-detectable oligoclonal bands, which are present in approximately 90% of MS patients [6]. In addition, follicular structures containing B lymphocytes have been found within the meninges of SPMS patients. The concept of the involvement of B lymphocytes in the pathogenesis of MS has resulted in the development of subsequent forms of pharmacological therapy, including the emergence of the first available drugs in the form of PPMS [7]. 

Thanks to the progress of pharmacotherapy in recent years, a significant slowdown in the progression and consolidation of neurological deficits in patients with MS has been achieved. Despite this, physiotherapy is still essential. New and more effective forms of rehabilitation are constantly being developed. In recent years, many studies have been conducted on the impact of various rehabilitation methods on improving both the performance and quality of life of MS patients [8]. It has been shown that the effectiveness of rehabilitation is associated with the phenomenon of so-called neuronal plasticity. Hence, attempts were made to determine the impact of rehabilitation on the stimulation of neuronal plasticity in patients with MS based on the expression levels of several markers in plasma, including neurotrophins and chemokines. Functional magnetic resonance imaging (fMRI) of the brain and assessment of the efficiency and clinical advancement of the disease also proved helpful [9]. 

Recent years have seen the development of telerehabilitation for remote exercises using immersive interfaces such as virtual reality (VR), augmented reality (AR), and mixed reality (MR) [10]. The possibility of exercises performed at home is particularly important in the case of mobility limitations of the patient, who can face problems with climbing stairs or getting to the rehabilitation site. 

Three-dimensional visualization technologies have gained particular popularity in recent years. Thanks to the development of digital techniques and information and communication technologies (ICT), the use of immersive interfaces is no longer just an element of entertainment but has become a new and exciting tool used in various fields of science, including medicine. Studies have examined the use of VR and AR in various conditions beginning with psychiatric disorders (depression, schizophrenia, autism, or drug disorders), neurological diseases (MS, after stroke, after spinal cord injuries, in Parkinson’s disease), and in the treatment of pain, as well as in physical rehabilitation and cognitive disorders [3,11,12,13]. The use of VR leads to the transfer of the user to a computer-generated reality, and the viewer has a sense of full presence in the virtually created world. The transfer to VR requires special head-mounted displays and data gloves, the use of which cuts the user off from the real world, and reduces the influence of external stimuli. AR employs electronic devices equipped with a camera, such as a smartphone or a tablet, and combines real-world images with computer-generated elements. MR is a hybrid reality in which the participant can interact with both real and virtual elements [10]. On the other hand, extended reality (XR) is a broad term covering all types of reality: VR, AR, MR, and technologies yet to be created, as shown in Figure 1.

## 2. Neuroplasticity–Molecular Basis

Brain plasticity is a key property of the nervous system allowing the formation of changes in response to stimuli from the body and environmental stimuli. The capacity for neuroplasticity is retained throughout life and is responsible for the ability to self-repair, learn, and adapt to emerging variables. It is a complex phenomenon caused by changes occurring at various levels of the nervous system’s organization. In the induction of neuroplasticity, the critical role is played by the bi-directional change in synapse strength, defined as synaptic plasticity, and expressed by long-term synaptic potentiation and long-term synaptic weakness [14]. Long-term potentiation (LTP) occurs as a result of an increase in the efficiency of synaptic transmission between simultaneously stimulated neurons. Changes in the amount and biophysical properties of AMPA-type glutamate receptors (AMPARs) in the postsynaptic membrane are responsible for the main mechanisms of induction of plastic changes in the central nervous system [15,16]. Among the best-known structures related to brain plasticity and LTP are the CA1 and CA3 areas of the hippocampus [17].

Multiple sclerosis is a heterogeneous disease in which different LTP and synaptic plasticity disorders are also observed in relapsing and progressive forms. In experimental autoimmune encephalomyelitis (EAE) in mice, synaptic transmission disorders have been shown to be based on an increase in glutamatergic transmission and a decrease in GABAergic transmission. In the case of MS patients, an analogous imbalance is observed, leading to a disturbed synaptic inhibition and excitation balance. In primary progressive multiple sclerosis (PPMS) patients, LTP is over-activated, which leads to excitotoxicity [18] and loss of the ability to induce synaptic plasticity, which translates into a clinical picture of the disease in which symptoms progress without the possibility of self-limitation. In the relapsing–remitting multiple sclerosis (RRMS) form, the ability to undergo plastic changes as a result of LTP induction remains preserved for a prolonged period of the disease, which results in the possibility of clinical improvement after relapse [19].

## 3. Markers of Neuroplasticity

Neuroplasticity is associated with changes in the levels of specific proteins called neuroplasticity markers in the blood and cerebrospinal fluid. A significant role in the induction of brain plasticity is played by neurotrophins: NGF, BDNF, NT-3, and NT 4/5. They are formed as precursor proteins that acquire biological activity in reaction with metalloproteinases [20]. Neurotrophins have a high affinity for tropomyosin-related kinase receptors and a low affinity for p75NTR receptors [21]. In addition, studies indicate that the hippocampus has an increased expression of TrkB and TrkC receptors, as well as the following neurotrophins: BDNF, NGF, and NT-3 [22]. In MS patients, neurotrophins may play a neuroprotective role either directly, by binding to Trk receptors on neurons in the central nervous system (CNS) and stimulating their survival, or indirectly, by binding to receptors on immune cells [21].

### 3.1. BDNF and Neuroplasticity

Among the neurotrophins, BDNF has the best-known mechanism of action and the most significant impact on the induction of brain plasticity. BDNF plays a fundamental role in the development and repair of the nervous system and its maintenance of function. In addition, it is the best-known neurotrophin acting through the mechanism of LTP induction. BDNF stimulates neuronal and axonal survival, plays a vital role in myelin repair, and stimulates neurogenesis in the dentate gyrus of the hippocampus [23]. In the case of MS patients, reduced levels of BDNF were shown in both the blood and the cerebrospinal fluid compared to those without organic CNS damage [24,25]. Two hypotheses have been put forward. Firstly, decreased BDNF levels lead to reduced CNS tissue protection in patients with MS, or secondly, these result from increased uptake by damaged nerve cells [26,27]. Serum BDNF levels are variable and increase during the flare and convalescence period compared to remission [28]. Plasma BDNF levels have also been found to be elevated in response to exercise training [21], and peripheral BDNF has the ability to cross the blood–brain barrier [29]. In addition, the activity of BDNF is stimulated by insulin-like growth factor-1. Under the influence of physical activity, there is an increase in the secretion of IGF-1 and VEGF, which, together with BDNF, lead to the development of synaptic plasticity by increasing the number and improving the functioning of synapses [30].

### 3.2. The Role of Myokines

Research is still underway to understand the exact mechanism that induces brain plasticity in response to rehabilitation. One of these mechanisms is the occurrence of the so-called muscle–brain endocrine loop [31], which proves that exercise increases the metabolic activity of muscle cells, which in turn induces increased production of myokines [32]. As muscle effector molecules, myokines act autocrine in muscle and exert endocrine and paracrine effects in other tissues by stimulating long-term synaptic potentiation underlying synaptic plasticity and neurogenesis. Thus, myokines influence structural and functional changes in the CNS in response to physical training [33]. For this reason, myokines may be considered potential biomarkers of brain plasticity, playing a role in the functioning of the whole organism. Their levels assessed in the blood or the cerebrospinal fluid may answer whether the type of rehabilitation intervention induces brain plasticity, as shown in Figure 2.

The first myokine that was identified was interleukin-6 (Il-6); it is released from muscles during their activity and influences other tissues’ metabolic processes [34]. It was found that the level of secreted Il-6 depends on the type of training, its intensity, and the duration of training [35]. In addition, Il-6 secreted during physical exercise by the muscles leads to the induction of the hypothalamic–pituitary–adrenal axis. This mechanism mediates the anti-inflammatory effect in contrast to monocyte-secreted Il-6 [36]. The anti-inflammatory effect of muscle Il-6 secreted during exercise occurs by inhibiting the production of TNF-alpha and stimulating the anti-inflammatory cytokines Il-1Ra and Il-10. In addition, muscle Il-6 is involved in stimulating glucose secretion in the liver, preventing the development of metabolic syndrome [31].

Another myokine that plays a role in exercise-induced brain plasticity is cathepsin B. It acts as a link between working muscles and the hippocampus. Cathepsin B stimulates the reconstruction of axons and increases the number of dendritic spines. It was found that the level of cathepsin B in the plasma increases during muscle activity. This is due to the activation of the AMP-activated protein kinase [37]. Cathepsin B activates matrix metalloproteinase 9 (MMP-9), which plays a role in extracellular matrix remodeling and modulation of synaptic plasticity [38]. In addition, cathepsin B has the ability to cross the blood–brain barrier, which increases the level of BDNF and doublecortin in the hippocampus, stimulating the processes of synaptic plasticity and neurogenesis [37]. This mechanism of action of cathepsin B may explain one of the crucial ways of inducing brain plasticity and may also explain the improvement of cognitive functions in response to physical training.

One of the myokines that induce brain plasticity in response to training is the adipomyokine irisin. Irisin is known mainly for the “browning” effect of white adipose tissue, and its effect on the stimulation of brain plasticity is associated with an indirect effect on the expression of BDNF in the CNS. Irisin is formed as a result of the cleavage of the membrane protein fibronectin type III domain-containing protein 5 (FNDC5). Physical activity increases the expression of peroxisome proliferator-activated receptor gamma coactivator 1 alpha (PGC-1 alpha), a transcriptional co-activator of mitochondrial biogenesis and a regulator of oxidative metabolism in brown adipose tissue and muscle [39]. PGC-1 alpha, on the other hand, increases irisin levels in the blood through the induction of membrane FNDC5 expression [40]. Subsequently, irisin crossing the blood–brain barrier induces BDNF expression in the hippocampus [41], enhancing brain plasticity.

### 3.3. Adipokines and Multiple Sclerosis

In addition to the involvement of muscle tissue in regulating brain plasticity, it is also worth considering adipose tissue and the adipokines it secretes. Adipokines are secreted by white adipose tissue and play a role in pro- and anti-inflammatory processes [42]. An unhealthy lifestyle can lead to obesity, causing an imbalance between secreted mediators from adipose tissue with a predominance of pro-inflammatory mediators. In turn, chronic inflammation leads to the activation of TLRs, which can cause immune system dysregulation observed in MS, among other conditions [43]. Differences are also encountered in the gut microbiome of obese patients [44]. Obesity has been found to play a role in the development of MS and is associated with a more severe disease course.

Among the adipokines that play a role in the pathogenesis of MS, we ought to mention adiponectin. It may participate in both anti-inflammatory and pro-inflammatory processes, probably depending on its isoform. Adiponectin levels in the blood and cerebrospinal fluid of MS patients have been found to be higher than in healthy individuals and correlate with a more severe course and faster progression of the disease [45,46]. Therefore, adiponectin could become a potential predictive biomarker of faster progression of MS in patients at the time of diagnosis. Physical activity leading to fat reduction could indirectly influence the inhibition of MS progression through the modulation of adiponectin [44]. Determining the levels of different adiponectin isoforms in MS patients in response to physical training could be an interesting direction for future research.

### 3.4. Oxidative Stress

Oxidative stress (OS) and nitroxidative stress (NOS) are considered to be significant factors in MS pathology. They occur as a result of the accumulation of reactive forms of oxygen and reactive forms of nitrogen, leading to mitochondrial dysfunction [47]. Mitochondrial dysfunction is responsible for axonal damage and leads to the development and progression of MS. OS is an exceedingly important factor that plays a role in the early stages of MS. In addition, OS has been found to negatively affect both the inflammatory and neurodegenerative components of the disease [48,49] through the oxidation of proteins, lipids, and DNA. Among the cells most sensitive to OS are oligodendrocytes and neurons, whose damage leads to demyelinating changes and axonal damage [50]. Furthermore, increased OS/NOS correlates with a more severe course of the disease, more relapses, and affects the development of depression in patients with MS [50]. 

By appropriately increasing physical activity, mitochondrial metabolism is improved [51]. However, it is important to remember that only properly selected training can benefit the patient. Unfortunately, the available data on the effects of different types of training on OS is partially disparate. Moderate-intensity aerobic exercise leads to a reduction in oxidative damage, while short-term high-intensity aerobic exercise can lead to an increase in oxidative damage. Attention should be paid not only to the intensity but also to the type of exercise. High-intensity cycling was found to reduce oxidative damage, while the same intensity of running increased it. Resistance exercise, on the other hand, caused a reduction in OS [52]. 

In the context of novel rehabilitation techniques using VR, there is a very limited base of work evaluating its effect on OS. A study evaluating chronic stroke patients who underwent training with VR found changes in OS t markers: there was a significant decrease in 8-hydroxy-2′-deoxyguanosine (8-OHdG) levels and a significant increase in heme oxygenase-1 (HO-1) after the intervention. Moreover, changes in OS marker levels were associated with improvements in patients’ functional outcomes [53]. Similar results were obtained in a group of patients undergoing conventional occupational therapy. The above results are encouraging and should prompt more researchers to evaluate the impact of VR techniques in regard to oxidative stress in patients with other diseases, including MS.

### 3.5. Epigenetic Inheritance

A different theory explaining the development of brain plasticity in response to physical activity is epigenetic inheritance, the molecular basis of which involves changes and regulation of gene expression without intervening in DNA sequence changes [54]. This leads to changes in the amount of protein encoded without changes in the genome. Epigenetic changes occur mainly through the processes of histone modification, DNA methylation, chromatin remodeling, and micro-RNA [55]. 

The primary mechanism responsible for regulating gene transcription is histone modification. It occurs through the activity of histone acetyl-transferase (HAT) and histone deacetylase (HDAC). Gene transcription can be activated by HAT and inhibited by HDAC [56]. Physical activity can increase HAT activity, leading to an increase in histone acetylation and a decrease in HDAC activity in the hippocampus [57]. Adaptive epigenetic changes arising in response to broad environmental changes lead to the induction of neuroplasticity in the hippocampus, mainly in the dentate gyrus. Consequently, this leads to an improvement in memory and cognitive function, as well as a reduction in anxiety disorders [58]. 

Another mechanism affecting gene expression through epigenetic inheritance is DNA methylation. It occurs as a result of the addition of a methyl group by DNA methyltransferase to the 5′cytosine carbon of the genomic cytosine–phosphate–guanine (CpG) dinucleotides. It has been found that both incidental and regular exercise can affect DNA methylation. In a mouse model, BDNF levels were found to be elevated after aerobic training, accompanied by a decrease in the expression of the neurotrophic factor receptor p75 [59]. Exercise-induced epigenetic chromatin remodeling can increase BDNF transcription and translation and affect BDNF gene chromatin remodeling [60].

### 3.6. The Mirror Neuron System

With the growing interest in using modern rehabilitation techniques using virtual reality, the question has arisen of whether and how these rehabilitation techniques affect the development of brain plasticity. The vast majority of studies evaluating the use of training with VR in patients with neurological conditions have found improvements in upper limb performance associated with increased strength and speed of movement [61,62,63]. Thus, we can try to hypothesize that VR training leads to improved motor performance in the course of brain plasticity induction. The exact mechanisms of such induction are currently unknown. One mechanism explaining the stimulation of neuroplasticity during VR exercise may be the induction of mirror neurons. The mirror neuron system (MNS) was described in 1996 by Rizzolatti et al. in oriental macaques in area F5 of the premotor cortex [64]. MNS activation occurs not only during the performance of motor activity but also during the observation of the performance of such activity by another person [65] in the case of the application of VR techniques by an avatar. The MNS in humans is depicted as a neural network with locations in both brain hemispheres within the inferior frontal cortex, the inferior parietal lobe, and the premotor cortex. 

The use of VR-based interventions leads to the activation of mirror neurons through multi-sensory feedback. As a result, areas of the sensorimotor cortex are activated, which translates into promoting and enhancing functional motor reorganization in the CNS [66].

Initially, researchers were enthusiastic about mirror neurons, but a group of skeptics questioned the existence and role of MNS in humans [67]. However, recent years have seen a renewed interest in mirror neurons in scientific papers. This may be due to the increased access to diagnostic tools, such as fMRI or transcranial magnetic stimulation (TMS), to assess MNS. 

## 4. Conventional Rehabilitation and Brain Plasticity

There is a substantial database of publications on the use of different types of rehabilitation in treating multiple sclerosis. It has been shown that rehabilitation in MS patients can lead to improvements in physical performance, balance, and mobility. In addition, rehabilitation improves the quality of life and reduces symptoms of fatigue and depression. Studies of conventional rehabilitation frequently involve evaluations of various plasma biomarkers in addition to the standard assessment of physical status. There is also an interesting trend in assessing the effects of rehabilitation on cognitive function changes. Thanks to the growing interest in the concept of brain plasticity, in recent years, researchers have also evaluated the effect of rehabilitation on its induction. Unfortunately, for innovative rehabilitation methods, there are limited reports evaluating the levels of biomarkers of brain plasticity in blood or cerebrospinal fluid. 

Among the available types of conventional rehabilitation, we do not have a clear answer as to which training can produce the best results in improving health in patients with MS (PwMS). One possible problem with this evaluation concerns the use of different research protocols regarding the duration, intensity, and frequency of interventions. Determining changes in the levels of not only clinical but also biochemical biomarkers can more clearly answer the question of what type of training is more effective. Regarding the plasticity markers, levels of peripheral BDNF, in response to aerobic training and combined exercise, were found to be significantly increased after the intervention. In addition, no differences were found between aerobic and combined training or the duration of the intervention on serum BDNF levels [68].

Moreover, chronic BDNF levels were also significantly elevated after rehabilitation. In contrast, exercise has not been shown to affect chronic NGF levels. However, a minor trend toward an increase in ciliary neurotrophic factor (CNTF) in response to exercise has been found [69]. 

It is well-known that BDNF plays a crucial role in inducing rehabilitation-induced brain plasticity. To identify ways to explain variability in the development of plasticity changes in response to the same inducers during neurorehabilitation, studies have searched for differences in BDNF polymorphism between patients. Among patients with MS, a correlation between atrophy of the brain’s gray matter and the BDNF Val66Met polymorphism has already been found. It was found that Met carriers demonstrated less atrophy of the gray matter and a smaller volume of demyelinating lesions in the T2 sequence [70]. Other studies have confirmed greater gray matter volume in Met carriers [71]. The effect of rehabilitation on improving the neurological status and performance in PPMS patients with the BDNF Val66Met polymorphism was also evaluated. PPMS patients who are Met carriers have been found to show significant improvement in the six-minute walking test (6MWT) compared to non-Met carriers in response to 3–4 weeks of inpatient neurorehabilitation [72]. The studies mentioned above may suggest that the BDNF Val66Met polymorphism may slow the progression of disability by activating correct patterns of cortical plasticity. The preceding topic represents an interesting strand of brain plasticity research and should be evaluated in future studies. If the current findings are confirmed in subsequent studies, the BDNF Val66Met polymorphism may become a prognostic biomarker for carriers, which may achieve a significant slowdown in disease progression due to the early implementation of neurorehabilitation. 

In comparison to RRMS forms, there are fewer papers on the evaluation of progressive forms of MS in terms of plasticity markers in the blood. One by Briken et al. assessed the effect of training on a bicycle ergometer on serum levels of irisin, IL-6, and BDNF in patients with progressive forms of MS. The study reported a significant increase in BDNF measured immediately after training, which decreased substantially after 30 min, even below pre-training baseline levels. No significant changes in myokine and BDNF levels were observed after a nine-week exercise cycle [73]. The lack of a significant increase in BDNF after rehabilitation in patients with progressive forms may indicate a dysfunction of the muscle–brain endocrine loop, which may lead to impaired induction of brain plasticity. The analysis mentioned above may attempt to explain the increasing disability in patients with progressive forms due to the depletion of the brain’s capacity for plasticity. The issue of the impact of physical rehabilitation on improving the efficiency of patients with progressive forms of MS remains unclear. It has been found that endurance training can lead to improved performance in these patients [74,75], but it can also lead to reduced depression, fatigue, and improved cognition [75]. On the other hand, there is a large database suggesting that patients with progressive forms do not significantly improve with physical rehabilitation [76].

Unfortunately, research confirms that MS patients are less physically active than healthy people. In addition, it was found that patients with PPSM, compared to RRMS patients, are less physically active [77]. 

## 5. Virtual Reality and Neurorehabilitation

Most of the studies conducted to date confirm the beneficial effect of virtual reality training on improving upper limb dexterity and on improving strength and speed of movement in patients with neurological conditions [61,62,63]. Unfortunately, most studies suffer from a small patient group size, which limits the value of the results obtained. A wide variety of tools related to 3D visualization techniques can be applied in rehabilitation. However, it is necessary to find the exact mechanisms affecting the neurological improvement of MS patients in response to different types of training using novel methods. Unfortunately, papers evaluating not only the effects of novel rehabilitation methods on improvements in physical performance or cognitive function [78] but also assessing plasma markers of plasticity, such as BDNF or myokines, are still awaited.

In a study using VR and a special upper limb motion controller (Leap Motion Controller), on a group of 16 patients with MS, a significant improvement in strength measured with a dynamometer was found for the hand stronger at the baseline. Compared to the control group, PwMS receiving traditional motor rehabilitation showed improved limb manual dexterity [79]. In a study comparing combined training VR + occupational therapy (OT) vs. OT, it was found that MS patients did not achieve significant differences in improving the agility of upper limbs in both groups. However, the patients using combined VR + OT training showed an improvement in the precision of hand movement. The results of the study suggested that adding VR exercises to traditional rehabilitation may have promising effects [80]. The limitation of the study was certainly the low size of the study group (*n* = 8) and the control group (*n* = 8). Another study comparing traditional training with video-based exergaming (VBE) in PwMS showed that both groups had a significant improvement in the agility of the upper limbs. In addition, a significant improvement in cognitive functions, balance, and reduction of fatigue was found. On the other hand, in the VBE group, the BDI scores decreased significantly [81]. Other interesting findings were provided by the study of the impact of unilateral upper limb VR training on the agility of the nontreated upper limb in MS patients. The more affected limb was subjected to VR training. It was found that after training with VR, the agility of the upper limbs, as assessed in 9-HPT, improved bilaterally, but a statistically significant improvement was found only in the exercised limb [82]. Improvement in the agility of an untrained limb may indicate the induction of plastic changes. The study points out that VR training can bring positive effects as a whole, not only within the part of the body involved in the training. However, results comparing VR techniques with traditional rehabilitation are contradictory [59,60,61] and more research in this area is required. On the other hand, interesting information on different types of rehabilitation, including VR rehabilitation, aerobic training, or yoga, was provided by the work of Hao et al. They evaluated the effectiveness of various rehabilitation methods among MS patients. Yoga, rehabilitation with VR, and aerobic exercise have been found to demonstrate better efficiency in improving balance, while training with VR, water exercise, and aerobic training lead to more significant improvement in walking impairments [83,84]. The use of augmented reality may also be associated with improved upper limb performance in MS patients. However, it was found that applied training with AR did not lead to changes in plasma BDNF or PDGF levels [85]. 

Despite the growing number of publications on rehabilitation using VR or AR in patients with neurological conditions, only a few researchers have addressed the effects on levels of plasticity biomarkers in blood or cerebrospinal fluid [63,64]. The mentioned assessment is needed to understand the mechanisms affecting neurological improvement through the induction of brain plasticity. 

It is worth noting that patient involvement plays a significant role in the rehabilitation process. Patients participating in traditional forms of rehabilitation frequently complain about their monotonous nature and, over time, become reluctant to participate in exercises. Moreover, it has been found that the level of patient involvement plays a crucial role in developing rehabilitation-induced neuroplasticity [86]. When innovative rehabilitation methods are applied, such as those based on virtual reality, a completely different aspect of rehabilitation options is presented to the patient. It was found that the use of telerehabilitation techniques in patients with neurological conditions in 2015–2019 was associated with increased patient involvement in the therapeutic process [87]. This is undoubtedly related to the creation of newer and more engaging forms of VR use in the rehabilitation process. 

## 6. Other Symptoms and Rehabilitation

In addition to motor symptoms, PwMS struggle with many other symptoms, such as cognitive decline, fatigue, and depressive disorders [2]. Fatigue symptoms are considered, in about 15% of PwMS, to be one of the key elements that reduce the quality of life [88]. Orexin is considered to be the main neurotransmitter responsible for fatigue symptoms in MS. It is secreted in the lateral hypothalamus. It is responsible for maintaining the circadian rhythm of sleep and wakefulness, and regulates appetite and metabolism [44]. The search for non-pharmacological methods of reducing fatigue is essential in the therapeutic process of MS, leading to attempts being made to use and evaluate new forms of rehabilitation in this area. Studies have confirmed the beneficial effects of traditional rehabilitation on reducing fatigue symptoms [89]. In addition, the use of VR in rehabilitation has also been found to lead to a reduction in fatigue and, at the same time, to an improvement in the quality of life of MS patients. Additionally, compared to traditional rehabilitation, VR rehabilitation is associated with better results in improving quality of life and reducing fatigue [3]. 

One of the significant accompanying symptoms of MS is ataxia. Unfortunately, pharmacological treatment brings little effect in reducing the symptoms of ataxia, which is why forms of non-pharmacological treatment of these symptoms are sought. The studies conducted so far do not show unequivocal results in terms of reducing the symptoms of ataxia in patients with MS [90,91,92]. The research also emphasized the issue that balance training should be combined with exercises stabilizing the appropriate section of the spine [93] or adding weight to the trunk [94]. An interesting issue is non-traditional forms of rehabilitation, such as ballet [95], which can lead to improved balance and reduced ataxia.

An essential role in the progression of disability in PwMS is played by cognitive impairment; it is found in more than 65% of patients and can occur at the early stages of the disease [2]. As there are currently no effective forms of pharmacological treatment for cognitive impairment [96], attempts have been made to evaluate the impact of rehabilitation on cognitive function in patients with MS. Unfortunately, recent studies have shown that traditional physical training in PwMS does not improve cognitive function both globally and concerning its specific domains [97]. The author recommends that his conclusions should be approached with caution, and more research in this area is needed. In terms of rehabilitation using VR in MS patients, improvements were found in executive and visuospatial functions, as well as in attentional and memory functions [98]. Comparing the impact of traditional rehabilitation vs. VR rehabilitation on cognitive impairment, it was found that both interventions improve mood and visual–cognitive functions. However, only the VR group showed improvements in learning ability and verbal short-term memory [78]. The presented results may prove that VR rehabilitation is an effective tool for improving cognitive functions in patients with MS. However, the results obtained require confirmation in further studies, in particular on large groups of patients.

## Figures and Tables

**Figure 1 jcm-12-01880-f001:**
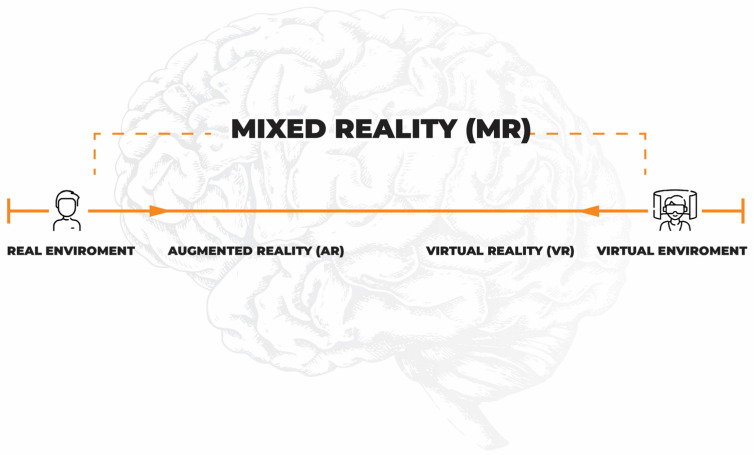
Visualization of immersive interfaces.

**Figure 2 jcm-12-01880-f002:**
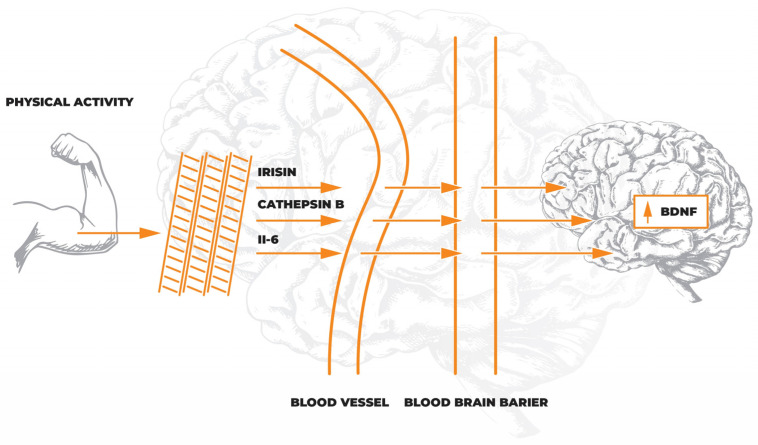
Diagram of the muscle–brain endocrine loop illustrating the process of induction of plastic changes in the brain in response to physical training. Physical activity leads to the release of myokines such as interleukin-6 (Il-6), irisin, and cathepsin B into the blood by striated muscles. By penetrating the blood–brain barrier, myokines induce an increased production of brain-derived neurotrophic factor (BDNF), mainly in the hippocampus. The presented process leads to improved neurogenesis and induction of neuroplasticity.

## Data Availability

The authors confirm that the data supporting the findings of this study are available within the article.

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
