# Peer review of "The Influence of Conventional and Innovative Rehabilitation Methods on Brain Plasticity Induction in Patients with Multiple Sclerosis"

_jcm, 2023, doi:10.3390/jcm12051880_

Round 1

Reviewer 1 Report

I can specify that the paper tries to give an overview of the rationale and utility of rehabilitation in PwMS, with a discussion about the new tecniques applied (e.g. VR, mirror neuron). This topic obviously is not original since the literature about rehabilitation and MS is abundant, however, for the same reason, a summary  could be considered helpful. I have only one advice for the authors, that is, to add in the discussion more data about clinical efficacy of a rehabilitation course on different clinical phenotypes of MS and data about different types of rehabilitation conducted on specific pattern of disability (limb weakness vs ataxia vs effect on fatigue).

Methods are appropriate and references are correctly uptodate.

Author Response

Thank you very much for your review and comments. We supplemented chapters: 4. Conventional rehabilitation and brain plasticity, 5. Virtual reality and neurorehabilitation and 6. Other symptoms and rehabilitation, with suggested case-control studies.

We have added, among others, data on rehabilitation in progressive forms, on the impact of rehabilitation on ataxia and cognitive functions.

Kind regards,

Authors

Reviewer 2 Report

The article "The Influence of Conventional and Innovative Rehabilitation 2 Methods on Brain Plasticity Induction in Patients with 3 Multiple Sclerosis" is interesting and could have implications in ameliorating the symptoms of MS in patients. The article is well organized and explains the different arenas of rehabilitation methods on brain plasticity. A few concerns need to be addressed:

1. Some similar recent articles are published recently such as https://doi.org/10.3390/jcm11237003, https://doi.org/10.3389/fnins.2021.707675. How is your article diiferent form those as many topics discussed are already in the literature.

2. The authors are encouraged to add case-control studies and explain the effect of the rehabilitation in MS, in addition to delienating the disease outcome, such as stratifying the degree of severity. 

3. The authors should also highlight a section on oxidative stress-related MS and Innovative Rehabilitation 2 Methods on Brain Plasticity Induction, to address the issue of neuronal oxidative stress which is common in MS.

4. Many recent refs could be cited: https://doi.org/10.1016/j.lfs.2018.12.029

5. Minor English corrections required for better clarity.

Author Response

Thank you very much for your review and comments.

With regard to point 1 - the presented manuscript concerns the phenomenon of neuroplasticity in relation to a group of patients with MS. In the work, we tried to present the current view on the basis of plasticity induction and available biomarkers of plasticity in the aspect of neurorehabilitation. In addition, in the work, we paid attention to new methods of rehabilitation, such as VR or AR, and tried to explain the potential of plasticity induction pathways using these methods. In this paper, we wanted a concise, but detailed presentation of the topic of neuroplasticity in MS. In addition, the manuscript is to encourage further researchers to conduct research in this field, with particular emphasis on directions that are assessed only in the basic scope (e.g. changes in biomarkers of brain plasticity in the blood and cerebrospinal fluid in response to VR training).

Of course, the MS-rehabilitation-neuroplasticity correlation has often appeared in recent years in publications. Nevertheless, in previous reviews, the authors focused mainly on traditional methods of rehabilitation. The subject of VR was very rarely taken up. Likewise, potential markers of neuroplasticity and their induction mechanisms have been rarely analysed. The presented work presents a combination of both these aspects.

Point 2 - We added more case-control studies to the discussion.

Point 3 - Thank you for drawing attention to the aspect of oxidative stress in MS. We've added a section on this topic. We also supplemented the data on the impact of traditional and innovative rehabilitation methods on the aspect of oxidative stress.

Point 4 - Citation has been added.

Point 5 - Proofreading of the manuscript was carried out by a native speaker.

Kind regards,

Authors

Reviewer 3 Report

The work has an interesting topic and is well written, nevertheless I have some things to suggest.

Specify in the abstract that it is a literature review.

Insert a paragraph specifying how the literature review was done.

Describe how the alteration of the immune system influences the onset of this pathology and which other systems it involves.

Among  nervous system mediators have an involvement in immune mechanisms but also in neuroplasticity (orexin)

How can lifestyle influence the onset of this pathology? How is adipose tissue involved? In this regard, the authors can refer to these works: 

DOI: 10.3390/ijms21239255

DOI:10.3389/fphys.2020.00356

Author Response

Thank you very much for your review and comments.

In response to comments:

1. Specify in the abstract that it is a literature review- Done

2. Insert a paragraph specifying how the literature review was done- We added a record that the latest literature was reviewed.

3. A paragraph describing the immunological basis of MS has been added.

4, 5, 6, 7- Thank you for your comments on the impact of adipose tissue and orexin on the pathology of MS. The manuscript has been supplemented with the above issues in the aspect of neuroplasticity and rehabilitation, taking into account the suggested citations.

Kind regards,

Authors

Round 2

Reviewer 3 Report

The authors reported all required changes.